# TRAINING-FREE AI-GENERATED IMAGE DETECTION VIA SPECTRAL ARTIFACTS

## ABSTRACT

The rapid progress of generative models has enabled the synthesis of photorealistic images that are often indistinguishable from real photographs, raising serious concerns about misinformation and malicious use. While most existing AI-generated image (AIGI) detection methods rely on supervised training with labeled synthetic data, they struggle to generalize to unseen generators and incur substantial overhead for retraining. In this work, we propose **SpAN**, a simple yet effective *training-free* detection framework based on spectral analysis. Our key observation is that upsampling operations in generative models inevitably introduce spectral artifacts, which remain most pronounced at the axial Nyquist frequencies, even when images appear realistic. Building on this insight, we design two techniques to enhance detection reliability: (1) power calibration via azimuthal integration to mitigate bias from image-specific frequency distributions, and (2) autoencoder-based reconstruction to amplify residual artifacts and enable discrepancy-based scoring between original and reconstructed images. Extensive experiments across multiple datasets and generative models demonstrate that SpAN achieves robust and generalizable detection performance. For example, SpAN outperforms other training-free detection methods by a substantial margin (+0.241 AUROC) in the Synthbuster benchmark, which contains recent generative models.

## 1 INTRODUCTION

Recent advances in generative models, including GANs (Huang et al., 2024) and diffusion models (Wang et al., 2024; Podell et al., 2023; Zhang et al., 2023; Zheng et al., 2023), have enabled the synthesis of highly realistic images that are often indistinguishable from real photographs. These *AI-generated images (AIGIs)* are now widely used for creative content generation (OpenAI, 2024), artistic design (Adobe, 2023), and educational support (Synthesia AI, 2023). However, they also raise serious concerns, such as deepfakes (Samantha Murphy Kelly, 2025), misinformation (Daniel Dale, 2025), and potential misuse in security-sensitive domains (Elizabeth Howcroft, 2025). As a result, reliable *detection of AIGIs* has become an urgent and important research problem.

Most existing AIGI detection methods rely on training-based detectors (Corvi et al., 2023; Karageorgiou et al., 2025; Dzanic et al., 2020; Chandrasegaran et al., 2021), where the detectors trained on a labeled binary classification dataset of real and AI-generated images, *e.g.*, ImageNet (Deng et al., 2009) *vs* Stable Diffusion (Rombach et al., 2022). While these methods have shown effective, they fundamentally suffer from several limitations: (*i*) they often fail to generalize to unseen generators or cross-domain scenarios (Jia et al., 2025), (*ii*), they require to collect AIGIs from diverse generators, and (*iii*) the training-based detectors must be frequently updated to remain effective. All these limitations could be problematic given the rapid development of new generative models.

To address these limitations, researchers have recently explored *training-free* approaches that detect AIGIs without relying on specific generative models or predefined real-image distributions (Ricker et al., 2024; He et al., 2024; Tsai et al., 2024; Brokman et al., 2025). For instance, AEROBLADE (Ricker et al., 2024) leverages reconstruction errors by passing an image through a pretrained autoencoder (*e.g.*, from Stable Diffusion), while RIGID (He et al., 2024) measures robustness to image perturbations in the latent embedding space of self-supervised models such as DINOv2 (Oquab et al., 2023). These approaches demonstrate the feasibility of AIGI detection without training. However, as generative models continue to improve (*e.g.*, producing high-resolution images with accurate

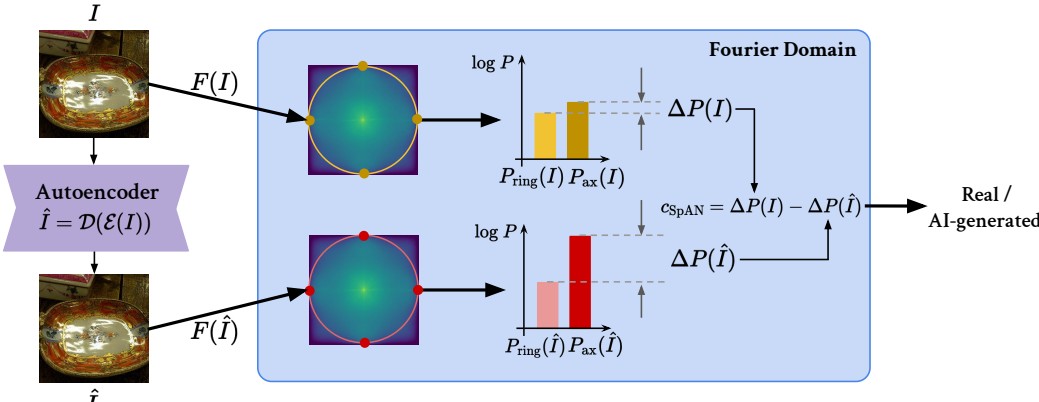

Figure 1: Overview of SpAN, the proposed training-free AIGI detection framework. SpAN first measures the power gap $\Delta P$ between $P_{\text{ax}}$, the power at axial Nyquist frequencies, and $P_{\text{ring}}$, the azimuthal integration of the high-frequency power. Then, the spectral discrepancy of $\Delta P$ between the original image $I$ and its reconstruction $\hat{I}$ is used for our criterion $c_{\text{SpAN}} = \Delta P(I) - \Delta P(\hat{I})$.

high-level semantics and realistic low-level details), visual cues become increasingly subtle and un-reliable. This motivates the following research question: *rather than searching for elusive signals in the image space, can we uncover systematic traces that persist even as image fidelity improves, for example in the Fourier domain?*

**Contribution.** In this paper, we mainly focus on *spectral artifacts* as a robust detection signal. It is well known that upsampling operations (*e.g.*, transposed convolutions) in generative models introduce checkerboard patterns in the Fourier domain (Karageorgiou et al., 2025; Zhang et al., 2019). In particular, the operations induce periodic replications in the power spectrum density, as shown in Figure 2. Although subsequent convolutional layers can reduce them, residual artifacts consistently remain at specific frequency locations. Our key observation is that these artifacts are most pronounced at the axial Nyquist frequencies, *i.e.*, $(\pm 0.5, 0)$ and $(0, \pm 0.5)$, because natural images typically concentrate most of their power near the zero frequency $(0, 0)$. Although there exist training-based methods that leverage frequency information for AIGI detection (Tan et al., 2024a; Jeong et al., 2022), the features that the models learn in the frequency domain may be limited to specific generator types (*e.g.*, GANs) (Corvi et al., 2023). Moreover, the use of the spectral artifact in a training-free AIGI detection has not been explored.

Based on our observation, we propose **SpAN**, a simple yet effective AIGI detection framework that leverages **Sp**ectral **A**rtifacts at **N**yquist frequencies of AI-generated images. Our key idea is to use the power at the axial Nyquist frequencies as the base criterion for detection. Since this can be biased by image content, we introduce two complementary techniques. First, we calibrate the criterion using azimuthal integration of high-frequency power, which mitigates bias from image-specific frequency distributions. Second, we exploit the discrepancy between the criterion computed on the original image and that on its autoencoder-based reconstruction, where the reconstruction process deliberately introduces artifacts, thereby allowing the original image's spectral characteristics to be assessed relatively. By integrating these steps, our final criterion becomes more robust and reliable. To the best of our knowledge, this work is the first to directly leverage spectral-domain information in the Fourier space as a metric for training-free AIGI detection. The overall framework is illustrated in Figure 1.

Extensive experiments demonstrate that our SpAN achieves state-of-the-art performance across standard AI-generated image detection benchmarks, Synthbuster (Bammey, 2023), GenImage (Zhu et al., 2023), and Chameleon (Yan et al., 2025) as reported in Table 1, 2, and 3, respectively. Notably, for high-resolution images (*i.e.*, Synthbuster), our SpAN outperforms the second best baseline by a large margin (**+0.241** AUROC). Furthermore, SpAN exhibits robustness over image corruptions compared to other baselines, as shown in Figure 4. These results highlights that spectral artifacts consistently exist across diverse generative models, even when AI-generated images appear photo-realistic. We believe our findings shed light on fundamental properties of generation models and can inspire future advances in the field of AI-generated image detection.

## 2 PRELIMINARIES

### 2.1 PROBLEM STATEMENT: TRAINING-FREE AI-GENERATED IMAGE DETECTION

We formulate *AI-generated image (AIGI) detection* as the task of defining a classification criterion that distinguishes between images synthesized by any generative model and real-world images captured from diverse sources (*e.g.*, cameras, digital drawings). Concretely, given an image $I \in \mathcal{I}$, our goal is to design a score function $c : \mathcal{I} \to \mathbb{R}$ that assigns higher values to AI-generated images $\mathcal{D}_{\mathrm{gen}}$ and lower values to real-world images $\mathcal{D}_{\mathrm{real}}$. In the standard evaluation practice, $\mathcal{D}_{\mathrm{real}}$ is sampled from a real dataset such as ImageNet (Deng et al., 2009), and $\mathcal{D}_{\mathrm{gen}}$ is constructed by a generative model, *e.g.*, Stable Diffusion (Rombach et al., 2022).

Most prior works (Karageorgiou et al., 2025; Wang et al., 2020; Tan et al., 2024b), adopt a training-based approach, using AI-generated images $\mathcal{D}_{\mathrm{gen}}$ from a specific generative model to learn the score $c(\cdot)$. While these methods have achieved strong detection performance, but they often fails to generalize to unseen generative models. Therefore, we mainly focus on a *training-free* setting where no prior information of $\mathcal{D}_{\mathrm{real}}$ and $\mathcal{D}_{\mathrm{gen}}$ is available in advance, and we aim to design a model-agnostic score $c(\cdot)$ that remains effective across diverse generative models.

### 2.2 FREQUENCY ANALYSIS OF IMAGES

In computer vision, frequency information provides a complementary perspective to spatial-domain representations, revealing structural patterns such as edges, textures, and periodic artifacts. These characteristics are often more easily captured in the frequency domain, making spectral analysis a powerful tool for image understanding and manipulation. Given an image $I$ of $H \times W$ pixels, its frequency representation can be obtained via the discrete Fourier transform (DFT):

$$F(u, v) = \sum_{x=0}^{W-1} \sum_{y=0}^{H-1} I(x, y) \cdot e^{-i2\pi(\frac{ux}{W} + \frac{vy}{H})},$$

where $(u, v)$ denote frequency coordinates. For convenience, the coordinates are often normalized to the range $[-0.5, 0.5]$. This normalization places the zero frequency at the center of the spectrum, with higher frequencies distributed toward the boundaries.

From the frequency coefficients $F$, one can compute the *power spectrum density (PSD)* as $P(u, v) = |F(u, v)|^2$, which quantifies the amount of power contained at each frequency. The PSD provides a concise characterization of the distribution of frequency components in the image, enabling analysis of whether most power is concentrated at low frequencies (*e.g.*, smooth variations) or high frequencies (*e.g.*, fine details or noise).

A key concept in spectral analysis is the *Nyquist frequency* $f_N$, defined as half of the sampling rate along each dimension. After coordinate normalization, this corresponds to the highest representable frequency at $u = \pm f_N$ and $v = \pm f_N$ where $f_N = 0.5$. Frequencies beyond this limit cannot be uniquely represented and are instead folded back into the base spectrum, a phenomenon known as aliasing. Formally, due to the periodicity of DFT, $F(u+1, v) = F(u, v)$ and $F(u, v+1) = F(u, v)$.

## 3 METHODOLOGY

In this section, we propose **SpAN**, a simple yet effective training-free AIGI detection framework using **Sp**ectral **A**rtifacts at **N**yquist frequencies of AI-generated images. To illustrate our framework, we first describe our observation of spectral artifacts at the axial Nyquist frequencies (Section 3.1). We then suggest a calibration technique for the artifacts to consider the amount of high-frequency information (Section 3.2). Finally, we design our detection criterion based on the spectral discrepancy between an input image and its reconstruction (Section 3.3). The overall framework is illustrated in Figure 1.

### 3.1 SPECTRAL ARTIFACTS AT AXIAL NYQUIST FREQUENCIES

We begin by describing our key observation on the spectral artifacts exhibited by generative models. It is widely known that a common artifact in images synthesized by generative models is the

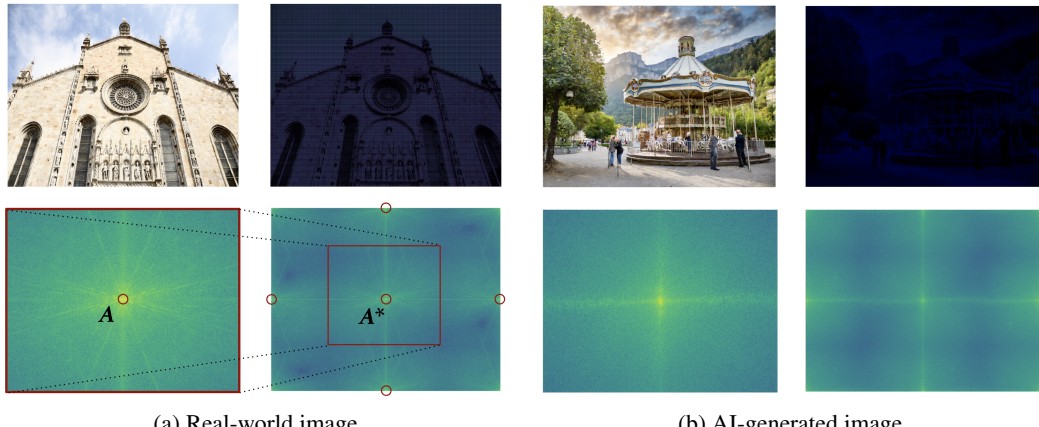

(a) Real-world image          (b) AI-generated image

Figure 2: Visualization of the emerging spectral artifact after applying single transposed convolution layer. Odd columns are the original images, and even columns are their upsampled versions, respectively. The lower row denotes the power spectrum density of an image at the identical column. The real-world image is sampled from RAISE-1k (Dang-Nguyen et al., 2015) and generated image is sampled from Firefly (Adobe, 2023).

appearance of *checkerboard patterns* in the frequency domain (Karageorgiou et al., 2025). Prior work has shown that these artifacts arise from the use of transposed convolutions of stride 2, where zeros are inserted in a "bed-of-nails" fashion during upsampling. This operation induces a periodic replication in the power spectrum density (PSD), as formally proven by Zhang et al. (2019).

Even when convolutional layers are subsequently applied, these artifacts do not fully vanish, especially at specific frequency regions. We find that the artifacts are most clearly preserved at the *axial Nyquist frequencies*, *i.e.*, $(\pm f_N, 0)$ and $(0, \pm f_N)$. This can be attributed to the fact that natural images typically exhibit their highest power near the zero frequency $(0, 0)$, and thus such upsampling operations also leads to relatively high power concentrated at the axial Nyquist frequencies.

For example, consider a real image $I \in \mathbb{R}^{C \times H \times W}$ and its PSD shown in Figure 2a. After upsampling $I$ to another image $I_{\text{up}} \in \mathbb{R}^{C \times 2H \times 2W}$ by a single transposed convolution as shown in Figure 2a, in the PSD of $I_{\text{up}}$, the midpoint of each edge of the spectrum acquires a significant power, the power of which is widely deviated from its adjacent region. In particular, the power of point $A$ (*i.e.*, $P(0, 0)$) in the original image $I$ is conveyed not only to the corresponding point $A^*$ in the upsampled image $I_{\text{up}}$, but also to midpoints of each side edge (*e.g.*, $P(-f_N, 0), P(f_N, 0)$), due to the periodic replication caused by the transposed convolution. Although the generated image in Figure 2b has exhibits fewer checkboard artifacts as it is generated through multiple convolution layers, significant power still remains along the central axis of the spectrum, including the axial Nyquist frequencies.

From this observation, one can expect that the power at the axial Nyquist frequencies is high for AI-generated images due to the use of transposed convolutions, while real images have a low power at the frequencies. Motivated by this, we propose to use the power as a simple criterion for AIGI detection, formally defined as:

$$P_{\text{ax}}(I) = \frac{1}{4} \sum_{(u,v) \in \{(\pm f_N, 0), (0 \pm f_N)\}} P(u, v),$$

where $P(u, v)$ denotes the PSD at frequency $(u, v)$ of the image $I$. In practice, we simply compute the average of nearest points of the axial Nyquist frequencies in the discrete PSD.

## 3.2 POWER CALIBRATION VIA AZIMUTHAL INTEGRATION

The distribution of frequency components may vary across images depending on their content, resulting in different amounts of high-frequency information and dominant frequency directions. Consequently, the power at the axial Nyquist frequencies $P_{\text{ax}}(I)$ may be biased by the amount of high-

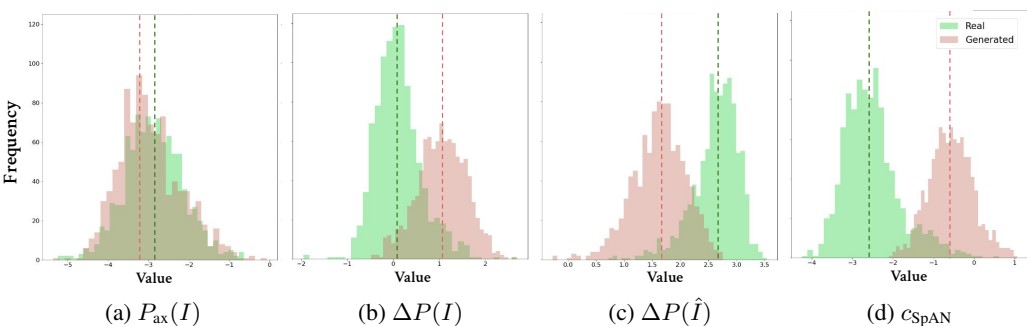

Figure 3: The distribution of $P_{\text{ax}}(I)$, $\Delta P(I)$, $\Delta P(\hat{I})$ and $c_{\text{SpAN}}$. For visualization, 1000 images are sampled from Midjourney and ImageNet in the GenImage benchmark, respectively. The vertical dashed-line denotes the mean value of each distribution.

frequency information. To calibrate this, we normalize $P_{\text{ax}}(I)$ using the azimuthal integration of power at the same frequency magnitude as follows:

$$\Delta P(I) = \log P_{\text{ax}}(I) - \log P_{\text{ring}}(I) \quad \text{where} \quad P_{\text{ring}}(I) = \frac{1}{2\pi} \int_0^{2\pi} P(f_N \cos\phi, f_N \sin\phi)d\phi.$$

To examine the efficacy of this calibration technique, we calcuate $P_{\text{ax}}(I)$ and $\Delta P(I)$ for 1,000 ImageNet (Deng et al., 2009) images and 1,000 Midjournery-generated (Midjourney Inc., 2023) images, and visualize their distributions in Figure 3a and 3b, respectively. Although $P_{\text{ax}}(I)$, the power at the Nyquist frequencies, is not a sufficient detection metric (see Figure 3a), its calibrated version $\Delta P$ provides much stronger discriminative power. These results highlight the importance of assessing how strongly certain artifacts appear relative to the overall frequency distribution, rather than relying solely on absolute power values.

In practice, the azimuthal integration is approximated by averaging the power over all frequency points that fall within a ring of width $\delta$ around the target magnitude as follows:

$$P_{\text{ring}}(I) = \frac{1}{|\mathcal{R}(f_N, \delta)|} \sum_{(u,v) \in \mathcal{R}(f_N, \delta)} P(u, v),$$

where $\mathcal{R}(r, \delta) = \{(u, v) : r - \delta \leq \sqrt{u^2 + v^2} < r\}$ denotes the set of frequency points that fall within the ring of radius $r$ and width $\delta$.

### 3.3 SPECTRAL ARTIFACT DETECTION WITH RECONSTRUCTION

The calibrated power $\Delta P(I)$ introduced in Section 3.2 may not be sufficient as a criterion when artifacts are relatively weak, such as in low-resolution generated images. To further enhance detection capability, we exploit the difference between an original image and its autoencoder-based reconstruction. The key idea is that the reconstruction can be regarded as an AI-generated image, since the autoencoder inevitably performs upsampling operations (*e.g.*, transposed convolutions) that generate grid-aligned spectral artifacts. Therefore, if the spectral discrepancy between the original and reconstructed images is large, the original is likely a real-world image because real images often have less artifacts; otherwise, it is likely to have been generated by a generative model.

Based on this intuition, we propose to use the discrepancy in the calibrated power $\Delta P(\cdot)$ between an original image $I$ and its reconstruction $\hat{I} = \mathcal{D}(\mathcal{E}(I))$, where $\mathcal{E}(\cdot)$ and $\mathcal{D}(\cdot)$ are the encoder and decoder of an autoencoder, respectively. Formally, our final detection criterion $c_{\text{SpAN}}(\cdot)$ is defined as follows:

$$\begin{aligned} c_{\text{SpAN}}(I) &= \Delta P(I) - \Delta P(\hat{I}) \\ &= \Big(\log P_{\text{ax}}(I) - \log P_{\text{ring}}(I)\Big) - \Big(\log P_{\text{ax}}(\hat{I}) - \log P_{\text{ring}}(\hat{I})\Big). \end{aligned}$$

This criterion is particularly effective for high-resolution images, which tend to exhibit stronger spectral artifacts due to multiple upsampling operations in a generation process. For low-resolution

images, we upsample the image $I$ while preserving its aspect ratio before feeding it into the autoencoder. This step ensures that the reconstruction process induces sufficient spectral artifacts, thereby making our discrepancy-based score a more reliable detection signal.

We further examine the effectiveness of this reconstruction-based technique by visualizing the distributions of $\Delta P(I)$, $\Delta P(\hat{I})$, and $\Delta P(I) - \Delta P(\hat{I})$ in Figure 3b-3d, respectively. For generated images, comparing the calibrated power of the original image $\Delta P(I)$ with that of the reconstructed image $\Delta P(\hat{I})$ shows little difference (*e.g.*, $\Delta P(I) \approx 1 \rightarrow \Delta P(\hat{I}) \approx 1.6$ in average), since artifacts are already present in the original. In contrast, for real images, new artifacts are introduced during reconstruction, leading to a significant increase (*e.g.*, $\Delta P(I) \approx 0 \rightarrow \Delta P(\hat{I}) \approx 2.5$ in average). Consequently, when examining the distribution of our final score $c_{\text{SpAN}} = \Delta P(I) - \Delta P(\hat{I})$, we observe substantially improved discriminative power.

## 4 Experiment

We design our experiments to validate the followings:

- Does our metric achieve strong performance in diverse AIGI detection tasks? (§4.1)
- How much does each component contribute to overall performance? (§4.2)
- Is our method robust over corruptions on the raw images? (§4.3)
- How does our method behave at the evaluation in the cross-domain setting? (§4.4)

**Evaluation Benchmarks.** We conduct evaluations on three widely used large-scale benchmarks in the field of AI-generated image detection: Synthbuster (Bammey, 2023), GenImage (Zhu et al., 2023), and Chameleon (Yan et al., 2025). The Synthbuster benchmark is composed of high-resolution images generated from 9 recent generative models, including propriety models, such as Firefly (Adobe, 2023), Midjourney (Midjourney Inc., 2023), DALL-E 2 (Ramesh et al., 2022), and DALL-E 3 (Ramesh et al., 2023), and open-sourced model like Stable Diffusion (Rombach et al., 2022). The real images are comprised of the subset of the RAISE dataset (Dang-Nguyen et al., 2015), which contains up to 4K resolution images. The GenImage benchmark contains relatively low-resolution images from 8 different generators. It includes images generated from GAN (Brock et al., 2018) and diffusion models where resolution ranges from $128 \times 128$ to $1024 \times 1024$, while the real-world images are the subset of ImageNet (Deng et al., 2009). Chameleon dataset comprises total of 26,033 images where real-images are collected from online communities for photographers, and generated images are based on community-tuned models and a proprietary model like DALL-E 3. More details about each benchmark can be found in Appendix C. The detection performance is measured by the area under ROC curve (AUROC).

**Implementation Details.** To implement our method, the ring width is set to $\delta = 0.01$, and Stable Diffusion 1.4 is used for the autoencoder. For small resolution images, we increase the image resolution so that the larger side of the image resolution becomes at least 1K. For a complete evaluation on Synthbuster and GenImage benchmarks, we used 2 NVIDIA RTX 4090 GPUs, each taking 3.5 and 40 hours, respectively.

**Baselines.** We compare the detection performance with recent training-free AIGI detection methods, RIGID (He et al., 2024), MINDER (Tsai et al., 2024), AEROBLADE (Ricker et al., 2024), and Manifold Bias (Brokman et al., 2025). To reproduce the result of AEROBLADE, we used Stable Diffusion 1.4 as the autoencoder, which is identical to our selection of the autoencoder. For Manifold Bias, we followed the official code provided by the authors that applies Stable Diffusion 2 as the latent diffusion model. In case of RIGID and MINDER, we used DINOv2 (Oquab et al., 2023) as the vision foundation model, and followed the threshold that is used to add noise or apply blurring as indicated in the paper.

### 4.1 Main Result

Tables 1, 2, and 3 are the evaluation results of our method and beselines on the Synthbuster, GenImage, and Chameleon benchmark. At the Synthbuster benchmark, our method exhibits the best performance among 4 other baselines achieving **+0.241** AUROC than the second best performing method. Also in the GenImage benchmark, our method exhibits the highest AUROC on average. While AEROBLADE performs competitive where the generative model matches the inspected autoencoder, it

Table 1: AI-generated image detection performance (AUROC) of proposed method and baselines in the Synthbuster benchmark (Bammey, 2023). We denote **bold** and underline as the best method and second best method, respectively.

| Method | Firefly | GLIDE | SDXL | SDv2 | SDv1.3 | SDv1.4 | DALL-E 3 | DALL-E 2 | Midjourney | Mean |
|---|---|---|---|---|---|---|---|---|---|---|
| RIGID | 0.519 | 0.868 | 0.757 | 0.615 | 0.448 | 0.446 | 0.442 | 0.596 | 0.593 | 0.587 |
| MINDER | 0.440 | 0.568 | 0.472 | 0.721 | 0.656 | 0.668 | 0.346 | 0.445 | 0.345 | 0.518 |
| AEROBLADE | 0.592 | **0.954** | 0.668 | 0.567 | 0.950 | 0.950 | 0.486 | 0.392 | 0.769 | 0.703 |
| Manifold Bias | 0.493 | 0.779 | 0.562 | 0.749 | 0.544 | 0.549 | 0.379 | 0.607 | 0.424 | 0.565 |
| SpAN (ours) | **0.945** | 0.893 | **0.988** | **0.948** | **0.994** | **0.994** | **0.948** | **0.795** | **0.989** | **0.944** |

Table 2: AI-generated image detection performance (AUROC) in the GenImage (Zhu et al., 2023) benchmark. We denote **bold** and underline as the best method and second-best method, respectively.

| Method | ADM | BigGAN | GLIDE | Midjourney | SDv1.4 | SDv1.5 | VQDM | Wukong | Mean |
|---|---|---|---|---|---|---|---|---|---|
| RIGID | **0.874** | 0.974 | 0.952 | 0.778 | 0.682 | 0.682 | **0.915** | 0.699 | 0.820 |
| MINDER | 0.768 | 0.681 | 0.582 | 0.450 | 0.607 | 0.596 | 0.882 | 0.676 | 0.655 |
| AEROBLADE | 0.856 | **0.981** | **0.989** | 0.918 | **0.982** | **0.984** | 0.732 | **0.983** | **0.928** |
| Manifold Bias | 0.727 | 0.925 | 0.852 | 0.510 | 0.675 | 0.673 | 0.874 | 0.653 | 0.736 |
| SpAN (ours) | 0.791 | 0.957 | 0.935 | **0.975** | 0.975 | 0.977 | 0.857 | 0.973 | **0.930** |

Table 3: AI-generated image detection performance (AUROC) in the Chameleon (Yan et al., 2025) benchmark. We denote **bold** and underline as the best method and second-best method, respectively.

| | RIGID | MINDER | AEROBLADE | Manifold Bias | SpAN (ours) |
|---|---|---|---|---|---|
| AUROC | 0.566 | 0.480 | 0.589 | 0.655 | **0.756** |

fails to generalize on the proprietary models such as DALL-E 2. Lastly, in the Chameleon benchmark, our method achieves +0.101 AUROC compared to the second best performing method. For the Chameleon benchmark, we did not report the performance per each model, because the specific type of the generator is not annotated to each image. We emphasize the performance in this benchmark as well, since it is composed of different generators, with more than 26k high-quality images. Overall, we could observe a notable gap between our method and recent compared methods especially when detecting recent AI-generated images, supporting the ability of generalization of SpAN to diverse generators.

## 4.2 ABLATION STUDY

**Component Ablation Study.** We observed the contribution of each component of our score $c_{\text{SpAN}}$, by sequentially adding each component suggested from §3.1 to §3.3. As shown in Table 6, purely using the averaged power at axial Nyquist points yield less discriminative result, because the absolute value may vary within both generated images and real-world images. However, by calibrating $P_{\text{ax}}$ by $P_{\text{ring}}$, we could achieve significant increase in performance, even without using any reconstruction process. As visualized in Figure 3b-3d and indicated in the third row of the Table 6, subtracting $\Delta P(\hat{I})$ widens the gap, initially observed at the distribution of $\Delta P(I)$ (see Figure 3d). This may be attributed to the fact that the reconstruction process 'cancels out' the artifacts of the original image, shifting the overall distribution of real-world images to the negative direction of the axis. Finally, by applying upsampling to original images before reconstruction, the value of $\Delta P(\hat{I})$ is intensified in case of real-world images, resulting in the best performance.

**Choice of Parameter.** We performed ablation on the parameter or architecture design to demonstrate that our method is not overly dependent on specific conditions. The width of the ring $\delta$ at $P_{\text{ring}}$ designates the broadness of a region that is used for calibration. We tracked the difference in the evaluation metric while increasing $\delta$ in the power of 2 from 0.01. As reported in Table 4, AUROC is preserved within the gap of 0.001 until $\delta = 0.08$, indicating the consistency of our method to size of the adjacent calibration region. Note that decreasing $\delta$ less than 0.01 makes it unavailable to define $P_{\text{ring}}$, as the width of the ring becomes too small for frequency points to fall within the region.

Table 4: Ablation on ring width $\delta$ at the Synthbuster benchmark. The best result is denoted in **bold**.

| $\delta = 0.16$ | 0.08 | 0.04 | 0.02 | 0.01 | 0.005 |
|---|---|---|---|---|---|
| 0.934 | 0.943 | 0.943 | 0.943 | **0.944** | N/A |

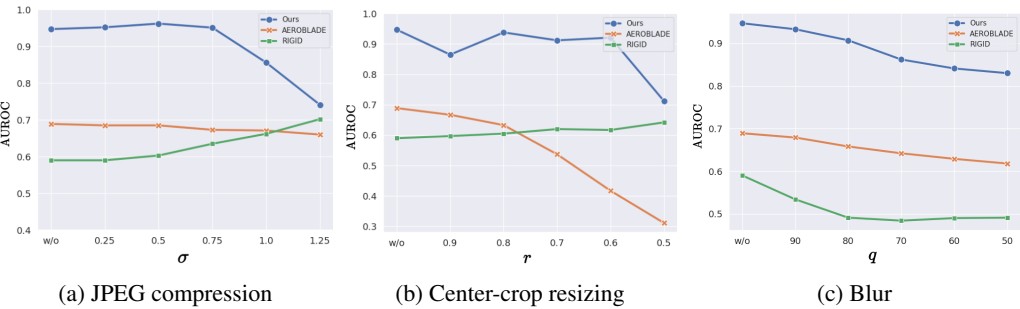

| (a) JPEG compression | (b) Center-crop resizing | (c) Blur |

Figure 4: Robustness to image corruptions of our method and baselines. For each corruption from (a) to (c), images are compressed by quality $q$, cropped by ratio $r$ and resized back to its original size, or blurred by standard deviation $\sigma$. Our method shows consistent superior result over other 2 baselines, validating its robustness to image perturbations.

Table 6: Component ablation study of our method on the GenImage benchmark. For component analysis, 1,000 images are sampled per each generative model with its corresponding real images. '✓' and '✗' denotes that the component is used, and not used respectively.

| Component | $P_{ax}(I)$ | $\Delta P(I)$ | $\Delta P - \Delta \hat{P}$ | AUROC |
|---|---|---|---|---|
| Nyquist Frequency (§3.1) | ✓ | ✗ | ✗ | 0.573 |
| + Power Calibration (§3.2) | ✓ | ✓ | ✗ | 0.849 |
| + Autoencoder-based reconstruction (§3.3) | ✓ | ✓ | ✓ | **0.930** |

**Choice of Autoencoder.** To compare the sensitiveness of our method to a specific autoencoder, we replaced the original Stable Diffusion 1.4 (SDv1.4) autoencoder with two different autoencoders, Stable Diffusion 2 (SDv2) and (Rombach et al., 2022) Kandinsky 2.1 (KDv2.1) (Arseniy Shakhmatov, 2023). Although SDv1.4 yields the best performance, replacing with other autoencoders still

Table 5: Ablation on autoencoder variants at the Synthbuster benchmark. The best result is denoted in **bold**.

| KDv2.1 | SDv2 | SDv1.4 |
|---|---|---|
| 0.857 | 0.860 | **0.944** |

outperformed other baseline methods, which supports invariance of our method to the choice of a specific model. This suggests that our method can benefit from common autoencoders that exhibit artifacts during upsampling in the generation process.

### 4.3 ROBUSTNESS TO CORRUPTIONS

For practical deployment in real-world scenarios, AIGI detectors should remain robust when applied to real-world usages that may undergo perturbations or postprocessing such as JPEG compression and image resizing. To evaluate this, we further assess the performance of SpAN on both real and AI-generated images under such perturbations. Specifically, we sample 500 real images from the RAISE-1k dataset (Dang-Nguyen et al., 2015) and 500 generated images from each model in the Synthbuster benchmark (Bammey, 2023). We then test three types of perturbations: JPEG compression, cropping and resizing, and Gaussian blurring, following (Ricker et al., 2024; Frank et al., 2020). The results of SpAN and the baselines are presented in Figure 4. As shown, SpAN maintains strong robustness and consistently outperforms the baselines even under the severe perturbation conditions.

### 4.4 COMPARISON OF CROSS-DOMAIN EVALUATION WITH TRAINING-BASED AIGI DETECTORS

We compare the performance of our method with frequency-aware, training-based AIGI detectors (Tan et al., 2024a; Karageorgiou et al., 2025) at the cross-domain dataset in a zero-shot setting, to support the necessity of training-free AIGI detection. For a fair com-

Table 7: Cross-domain evaluation result of proposed method and baselines. The best result is denoted in **bold**.

| | FreqNet | SPAI | SpAN (ours) |
|---|---|---|---|
| AUROC | 0.604 | 0.478 | **0.758** |

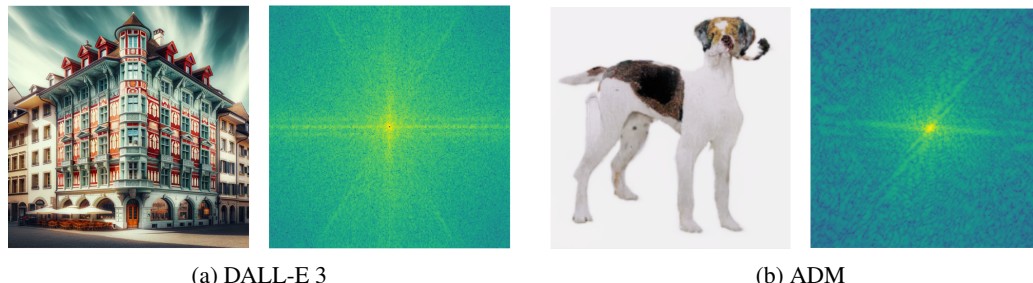

(a) DALL-E 3              (b) ADM

Figure 5: A set of generated image and its power spectrum density map each sampled from DALL-E 3 in the Synthbuster benchmark, (Ramesh et al., 2023) and ADM (Dhariwal & Nichol, 2021) in the GenImage benchmark.

parison, we chose WikiArt[1] dataset from the art domain, where both the pre-trained autoencoder of our method and the domain-specific trained detectors have been rarely exposed to. As shown in Table 7, our method performs better than both training-based AIGI detectors at the zero-shot inference on the unseen data. This experiment demonstrates the weakness of training-based AIGI methods of being prone to detecting out-of-distribution samples, although the detectors might have classified images accurately at the learned data manifold. Conversely, our method focuses on upsampling artifact that is not severely impacted by the distribution shift of the inference data, therefore could preserve the performance.

### 4.5 CASE STUDY

In this subsection, we show specific cases of how our method can behave according to the characteristics of the generated images in the Fourier domain. Figure 5 shows a generated image from DALL-E 3 (Ramesh et al., 2023) and ADM (Dhariwal & Nichol, 2021), and its counterpart converted by the discrete Fourier transform, respectively. For 5a, which is composed of high-frequency details, our method can behave better by capturing the artifact from the raw state of the generative image, resulting in a relatively high $c_{\text{SpAN}}$. This is mainly because $\Delta P(I)$ is big enough to cancel the effect of subtracting $\Delta P(\tilde{I})$. In contrast, an over-blurry image, such as in Figure 5b, can unintentionally mimic the distribution of a real-world image in the Fourier domain, which paradoxically becomes relatively difficult to detect. However, considering that recent generative models are becoming closer to real-world images imitating high-frequency details, this property can become an advantage in the near future. Also, compared to the previous reconstruction-based model, which has the assumption that generated images are harder to reconstruct, our method has the strength to handle high complexity images by observing the artifact of the upsampling process.

## 5 RELATED WORKS

### 5.1 TRAINING-FREE AI-GENERATED IMAGE DETECTION

To address the rapid proliferation of generative models, training-free detection methods which do not require AIGIs for training have recently emerged. Most existing approahces leverage the pre-trained representations of large foundation models (*e.g.*, DINOv2 (Oquab et al., 2023)) for detection. For instance, Ricker et al. (2024) measures the perceptual distance between an original image and its reconstruction by the Latent Diffusion Model (LDM) autoencoder, based on the observation that images generated by LDMs exhibit lower reconstruction error when evaluated by the corresponding LDM. On the other hand, He et al. (2024) and Tsai et al. (2024) exploit the robustness of self-supervised vision foundation models to perturbations like Gaussian noise or blurring, under the hypothesis that real images are inherently more robust to such distortions. Brokman et al. (2025) assumes that real data are more likely to reside on the latent-space manifold of the LDM. In contrast to these approaches, which primarily depend on predictions from pre-trained models, we demon-

---
[1] https://www.wikiart.org/

strate that image-specific frequency information remains highly effective for detecting AIGIs in a training-free regime.

## 5.2 AI-GENERATED IMAGE DETECTION VIA FREQUENCY ANALYSIS

Several training-based AIGI methods have leveraged frequency information as the key representations, as generated images tend to exhibit various artifacts in the frequency domain (Li et al., 2024; Dzanic et al., 2020; Chandrasegaran et al., 2021). Durall et al. (2020) pointed out the spectral distortion in the images generated from the CNN-based model and utilized the gap to detect deep-fake images. Tan et al. (2024a) leveraged the high-frequency characteristics in GAN-based generators. Frank et al. (2020) investigated artifacts in GAN-generated images in the frequency domain by applying the discrete cosine transform (DCT), and indicated the artifact as a result of upsampling techniques. Another approach is to learn a deepfake detector with a perturbation generator as in Jeong et al. (2022). Karageorgiou et al. (2025) utilized masked spectral learning to learn the spectral distribution of real images, considering AI-generated images as out-of-distribution samples. Although analysis based on the Fourier domain has been used as a distinctive factor for discriminating generated images, this learned frequency distribution may be subject to a specific domain or dataset, leading a performance degradation in out-of-distribution data.

## 6 CONCLUSION

In this work, we propose SpAN, a simple yet effective training-free AIGI detection method inspired by the spectral artifacts of generated images observed in the Fourier domain. By comparing the energy gap near the axial Nyquist frequency before and after image reconstruction, we could robustly discriminate AI-generated images from real-world images. Extensive experiments demonstrate the effectiveness of our framework across diverse benchmarks and types of generative models, as well as its robustness to image perturbations. We hope that our research will be expanded to exploiting other artifacts residing in generated images in the training-free setting of AIGI detection.

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

## A  PSEUDOCODE OF SPAN

---

**Algorithm 1** SpAN (PyTorch-like Pseudo-code)

---

```
# AE   : autoencoder
# DFT  : discrete Fourier transform function
# RING : set of frequency points within the ring

def compute_power(I):
    PSD = DFT(I)
    P_ax = torch.mean([PSD[-0.5, 0], PSD[0.5, 0], PSD[0, -0.5], PSD[0, 0.5]])
    P_ring = torch.mean([PSD[i, j] for i, j in RING])
    return P_ax, P_ring

def SpAN(I):
    I_hat = AE(I)
    P_ax, P_ring = compute_power(I)
    P_ax_hat, P_ring_hat = compute_power(I_hat)
    return torch.log(p_ax / p_ring) - torch.log(p_ax_hat / P_ring_hat)
```

---

We outline the high-level algorithm of SpAN in a PyTorch-like pseudo-code format in Algorithm 1.

## B  FURTHER EXPERIMENTAL DETAILS

We describe additional implementation details such as preprocessing step in our method and the settings we utilized when reproducing the result of baselines.

**Additional Implementation Details.** At the preprcoessing step of our method, raw images are converted from RGB to grayscale images and then subtracted by the mean of the entire image pixel values, to effectively capture the spectral artifacts. We utilize the Hann window function (Harris, 2005) to reduce spectral leakage, followed by the fast Fourier transform (Welch, 2003) as the discrete Fourier transform function. Stable Diffusion 1.4 (Rombach et al., 2022) is utilized as our autoencoder, where the pre-trained weight is provided from the Huggingface[2]. To accurately reflect the true spectral distribution of the raw image, the power spectrum density of a reconstructed image is compared with that of the original image.

**Baselines.** For reproducing the results of AEROBLADE and Manifold Bias, we followed the default setting of its official github repository. For AEROBLADE, we report the performance when the input image resolution is constrained to the fixed-sized as stated in the paper's supplementary material. For RIGID and MINDER, images are resized to their default value by $224 \times 224$, where the DINOv2 model with backbone of ViT/L-14 (Dosovitskiy, 2020) is utilized as the vision foundation model.

## C  INFORMATION OF EVALUATION BENCHMARKS

We present the metadata of the three benchmarks that are used for the evaluation, Synthbuster (Bammey, 2023), GenImage (Zhu et al., 2023), and Chameleon (Yan et al., 2025) at Table 8, 9, and 10, respectively. For a dataset that is comprised by images of various resolutions, we simply notate its resolution in ranges (*e.g.*, 1K $-$ 2K) instead of fixed resolutions (*e.g.*, $512 \times 512$). We also summarize the characteristic of each benchmark in the followings:

- **Synthbuster benchmark** (Bammey, 2023) contains high quality images where real-world images are a subset of RAISE (Dang-Nguyen et al., 2015) and generated images are collected from recent generators. Specifically, all generators are published from 2022 except for GLIDE (Nichol et al., 2021). It contains images produced by various diffusion-based models and commercial APIs.

- **GenImage benchmark** (Zhu et al., 2023) is made up of real-world dataset from the ImageNet (Deng et al., 2009) subset and generated dataset from 8 different generative models, consisting

---

[2]https://huggingface.co/CompVis/stable-diffusion-v1-4

of a total of 50k+50k samples. The dataset is divided into 6k or 8k per generator, and the number of real-world images is matched with the corresponding generator.

- **Chameleon benchmark** (Yan et al., 2025) also contains high quality images where the resolution of the images is ranged from 720p to 4K. Both real-world images and generated images are annotated by 4 major categories (*i.e.*, 'scene', 'object', 'animal', 'human'). In particular, data curation is applied to ensure the collection of high quality generated images by going through a 'perception turing test'. The real-world images are collected from online communities for photographers such as Unsplash (uns).

| | Dataset | Resolution | Dataset size | Generator type |
|---|---|---|---|---|
| **Real-world** | RAISE-1k (Dang-Nguyen et al., 2015) | $1K - 4K$ | 1k | - |
| | Firefly (Adobe, 2023) | $2K - 4K$ | 1k | Commercial API |
| | GLIDE (Nichol et al., 2021) | $256 \times 256$ | 1k | Diffusion model |
| | SD v1.3 (Rombach et al., 2022) | $512 \times 512$ | 1k | LDM |
| | SD v1.4 (Rombach et al., 2022) | $512 \times 512$ | 1k | LDM |
| **AI-generated** | SDXL (Podell et al., 2023) | $1K - 2K$ | 1k | LDM |
| | SD v2 (Rombach et al., 2022) | $1K - 2K$ | 1k | LDM |
| | DALL-E 2 (Ramesh et al., 2022) | $1024 \times 1024$ | 1k | Diffusion model |
| | DALL-E 3 (Ramesh et al., 2023) | $1K - 2K$ | 1k | DiT |
| | Midjourney (Midjourney Inc., 2023) | $1K - 2K$ | 1k | Commercial API |

Table 8: Metadata of the Synthbuster (Bammey, 2023) benchmark

| | Dataset | Resolution | Dataset size | Generator type |
|---|---|---|---|---|
| **Real-world** | ImageNet (Deng et al., 2009) | $300 - 1000$ px | 50k | - |
| | ADM (Dhariwal & Nichol, 2021) | $256 \times 256$ | 6k | Diffusion model |
| | BigGAN (Brock et al., 2018) | $128 \times 128$ | 6k | GAN |
| | GLIDE (Nichol et al., 2021) | $256 \times 256$ | 6k | Diffusion model |
| **AI-generated** | Midjourney (Midjourney Inc., 2023) | $1024 \times 1024$ | 6k | Commercial API |
| | SD v1.4 (Rombach et al., 2022) | $512 \times 512$ | 6k | LDM |
| | SD v1.5 (Rombach et al., 2022) | $512 \times 512$ | 8k | LDM |
| | VQDM (Gu et al., 2022) | $256 \times 256$ | 6k | Diffusion model |
| | Wukong (wuk, 2022) | $512 \times 512$ | 6k | Diffusion model |

Table 9: Metadata of the GenImage (Zhu et al., 2023) benchmark

| | Dataset | Resolution | Dataset size | Generator type |
|---|---|---|---|---|
| **Real-world** | Unsplash (uns) | $720P - 4K$ | 14,863 | - |
| **AI-generated** | Community-tuned models, DALL-E 3, Stable Diffusion, etc. | $720P - 4K$ | 11,170 | various |

Table 10: Metadata of the Chameleon (Yan et al., 2025) benchmark

