# OpenReview forum: "Training-free AI-generated Image Detection via Spectral Artifacts"
_ICLR.cc/2026/Conference — Submitted to ICLR 2026_

### Official Review · Reviewer_Q8Fx · 2025-10-30

**Soundness:** 2
**Presentation:** 2
**Contribution:** 2
**Rating:** 4
**Confidence:** 3

**Summary:**

This paper proposes SpAN, which is a training-free detector for AI-generated images based on spectral analysis of artifacts. The method measures the axial Nyquist power in an image and calibrates it by comparing it to the power through azimuthal integration. The image is passed through a pretrained autoencoder  and the calibrated power of the original and reconstructed image is compared. The paper showed experiments on two benchmarks and claimed that SpAN achieves higher performance compared to existing methods.

**Strengths:**

1. The paper introduces a training-free detection method, which is also easily adaptable and efficient to deploy for real-world scenarios.
2. The method utilizes the discrepancy between the original image and a generated image through an autoencoder, to assess the spectral characteristics relative to the original image. This is a dynamic setup, which should help in the generalization of the method.

**Weaknesses:**

1. The paper's methodology stems from the fact that generative models exhibit spectral artifacts, which arise from transposed convolutions of stride 2. However, there are generative models that don't utilize transposed convolutions. There is no ablation study regarding this. If this method only works for models that have transposed convolutions, then this will be a severe limitation of the methodology.
2. Figure 3 shows that there is indeed separability between generated and original images. However there is overlap in the distributions too, making this method sensitive to specific images. I would be interested to know how this method performs when a real and generated image have very similar visuals and features. Furthermore, if an image is generated using inpainting, how would the performance be effected? I would imagine this method will be sensitive to those images.
3. The evaluation shows performance in two benchmarks Synthbuster and GenImage.  The method shows good performance in Synthbuster, but is rarely the best model in GenImage. The paper didn't try to explore this further, and there is no mention why the performance might have degraded. The authors should benchmark on more datasets (for e.g. [1]) to show proper performance comparison.
4. Due to the nature of the method, I would also be very interested in how often the model misclassifies real images. I would suggest to include ablation studies by taking real images from ImageNet, COCO, etc, and show the misclassification scores.

[1] Rahman, Md Awsafur, et al. "Artifact: A large-scale dataset with artificial and factual images for generalizable and robust synthetic image detection." 2023 IEEE International Conference on Image Processing (ICIP). IEEE, 2023.

**Questions:**

1. How would the method perform on images from a generator that does not use explicit transposed convolution layers? Is there any experiment the authors have done regarding this?
2. Did the authors do experiments that show methodology performance when generated images are very similar to real images, or which is generated through inpainting?
3. Do the authors show performance on extended benchmarks to show robustness of their methodology?
4. Can the authors show performance specific to real images?

---

> ### Author Response · Authors · 2025-11-21
> **Rebuttal by Authors [1/4]**
>
> Dear Reviewer Q8Fx,
>
> We thank the reviewer for the thoughtful and helpful remarks. We respond to the questions and concerns one-by-one.
>
> ---
>
> > **[Q1]** How would the method perform on images from a generator that does not use explicit transposed convolution layers? Is there any experiment the authors have done regarding this?
>
> **[Response 1]**
> Generative models without explicit transposed convolution layers are already included in our evaluation benchmarks (e.g., Stable Diffusion XL [1], GLIDE [2], DALL-E 2 [3]), and SpAN could effectively detect those images as shown in Table 1 and 2 in the paper.
>
>
> In our paper, the transposed convolution [4], which is a single learnable linear operator inserting zero-valued pixels and applying a learnable convolution filter, is utilized as one example of demonstrating the artifact that can arise due to the upsampling operation.
>
>
> Meanwhile, non-transposed convolution upsampling, such as applying interpolation for upscaling the image and applying normal convolutional layer afterwards are included in recent generative models (e.g., Stable Diffusion XL [1]). Moreover, GLIDE [2]  and DALL-E 2 [3] are based on a super-resolution Diffusion model, where their U-Nets contain non-transposed convolution upsampling layers for feature maps.
>
> Although these advanced upsampling strategies might generate smoother artifacts near the Nyquist frequency, as shown in Table 1, 2, SpAN could still notice the spectral artifact, where power calibration or the reconstruction process can become more helpful.
>
> In short, SpAN prioritizes the _‘existence’_ of an upsampling process rather than being dependent on a _‘specific type’_ of an upsampling process. This is possible because artifacts at the axial Nyquist frequencies are mainly generated by the periodic frequency replicas (see Figure 2a), which is less dependent on a certain type of the upsampling operation.
>
> This characteristic of SpAN can be advantageous especially for detecting recent high-quality images generated from proprietary models (e.g., Midjourney, Firefly), as these models usually require repetitive upsampling for generating high resolution images.
>
> **References**
>
> [1] Podell, Dustin, et al., "Sdxl: Improving latent diffusion models for high-resolution image synthesis.", arXiv preprint 2023. \
> [2] Ramesh, Aditya, et al., "Hierarchical text-conditional image generation with clip latents.", arXiv preprint 2022. \
> [3] Nichol, Alex, et al. "Glide: Towards photorealistic image generation and editing with text-guided diffusion models." arXiv preprint 2021. \
> [4] Odena, Augustus, Vincent Dumoulin, and Chris Olah., "Deconvolution and checkerboard artifacts.", Distill 2016.

---

> ### Author Response · Authors · 2025-11-21
> **Rebuttal by Authors [2/4]**
>
> > **[W2]** I would be interested to know how this method performs when a real and generated image have very similar visuals and features. Furthermore, if an image is generated using inpainting, how would the performance be effected? I would imagine this method will be sensitive to those images.
> **[Q2]** Did the authors do experiments that show methodology performance when generated images are very similar to real images, or which are generated through inpainting?
>
>
> **[Response 2]**
> The artifact near Nyquist frequency originates from the upsampling process of generative models, so it is less likely that our method be sensitive to visually or semantically similar image pairs. For validation, we manually chose 100+100 real and generated toy dataset from Chameleon benchmark [1] (Please see Response 3 for the details for the Chameleon benchmark). Among the dataset which would likely be classified as ‘human’ in the Chameleon benchmark, we sorted out where the face of the person is captured in the images, to match the visual similarity between real-world and generated images.
>
> SpAN achieves AUROC=0.784 for the face subset while achieving AUROC=0.756 for the entire Chameleon dataset. This result suggests that even though the generated images have similar visuals with real-world images, SpAN can robustly identify generated images by focusing low-level information in the Fourier domain.
>
> Furthermore, we have conducted an experiment on detecting images that are inpainted from real-world images. We here use Stable Diffusion 2 inpainting data used in [2], which is generated by the official Stable Diffusion 2.1 repository [3]. Please note that background regions that are excluded for inpainting are replaced by the original background of the corresponding real-world image, resulting in the mixed property between real-world image and generated images.
>
> |  | RIGID | MINDER | AEROBLADE | Manifold Bias | SpAN (ours) |
> | :---- | :---- | :---- | :---- | :---- | :---- |
> | AUROC | 0.604 | 0.605 | 0.613 | 0.523 | 0.578 |
>
> While SpAN shows relatively low performance compared to the case of detecting raw generated images, other training-free baselines also tend to suffer achieving AUROC between 0.523~0.613.
> For SpAN, this may be due to the fact that if the inpainting regions take a small portion of the original image, the spectral artifact can become harder to detect, as the impact of spectrum from the real-world image gets stronger, suppressing the artifacts.
>
> Although inpainting images were not the main consideration in the training-free detection of AIGI setting, it would be an interesting research direction to investigate the artifacts of generated images after splitting raw images into smaller patches, dedicated for detecting inpainting images. We leave this direction as a room for our future work.
>
> **References**
>
> [1] Shilin Yan, et al.,” A sanity check for ai-generated image detection”, ICLR 2025. \
> [2] Guillaro, Fabrizio, et al., "A bias-free training paradigm for more general ai-generated image detection.", CVPR 2025. \
> [3] Stable Diffusion official github repository, “https://github.com/Stability-AI/stablediffusion”

---

> ### Author Response · Authors · 2025-11-21
> **Rebuttal by Authors [3/4]**
>
> > **[W3]** The method shows good performance in Synthbuster, but is rarely the best model in GenImage. The paper didn't try to explore this further, and there is no mention why the performance might have degraded. The authors should benchmark on more datasets to show proper performance comparison.
> **[Q3]** Do the authors show performance on extended benchmarks to show robustness of their methodology?
>
> **[Response 3]**
> To resolve your concern, we have conducted additional evaluation on another widely used benchmark ‘Chameleon’ [1]. We chose Chameleon as the additional benchmark since it includes up-to-date generators with proprietary models, which is more widely used today.
>
> |  | RIGID | MINDER | AEROBLADE | Manifold Bias | SpAN (ours) |
> | :---- | :---- | :---- | :---- | :---- | :---- |
> | AUROC | 0.566 | 0.480 | 0.589 | 0.655 | **0.756** |
>
> Our method significantly outperforms existing training-free methods in the Chameleon dataset, which comprises 26,033 images in total. We emphasize that Chameleon is a large-scale benchmark that contains highly realistic generative images from several different generators. This result supports the superior performance of our method compared to the baselines. Also, please refer to Appendix C for more details about the Chameleon benchmark.
>
> Also, we explained the possible reason for performance degradation at the GenImage benchmark in the ‘Case Study’ (see section 4.5 and Figure 5b). As stated in the paper, generated images that are over-blurry can be ambiguous for SpAN as the distribution of the PSD might be similar to real-world images. However, this phenomenon is rarely observed in recently generated images, as latest generative models are more likely to mimic the true distribution of real-world images which contain high frequency details.
>
> To verify the generalization of SpAN to images that are generated from recent models, we have conducted an evaluation on 4 more dataset generated from FLUX1.0-dev [3] (2024), Stable Diffusion v3 [4] (2024), GigaGAN [5] (2023), and Midjourney v6 [6] (2024). We use the dataset subset that is used in SPAI [8] as other models are already included in our benchmarks. The real-world image is ImageNet from the GenImage benchmark [7].
>
> |  | FLUX \[3\] | SDv3 \[4\] | GigaGAN \[5\] | MJv6 \[6\] | Mean |
> | :---- | :---- | :---- | :---- | :---- | :---- |
> | RIGID | 0.955 | 0.761 | 0.872 | &nbsp;&nbsp;0.890 | 0.870 |
> | MINDER | 0.599 | 0.599 | 0.633 | &nbsp;&nbsp;0.665 | 0.624 |
> | AEROBLADE | 0.771 | 0.619 | 0.789 | &nbsp;&nbsp;0.667 | 0.712 |
> | Manifold Bias | 0.443 | 0.340 | 0.359 | &nbsp;&nbsp;0.496 | 0.410 |
> | SpAN (ours) | **0.985** | **0.990** | **0.996** | &nbsp;&nbsp;**0.970** | **0.987** |
>
>
> As shown in the above table, our method achieves best performance on every recent model, with +0.117 AUROC compared to the second best performing method. This result validates the strength of SpAN to recent models. Moreover, please note that our method can detect generated images from various types of generators including recent GAN (GigaGAN) or flow-matching based model (FLUX) .
>
> **References** \
> [1] Shilin Yan, et al.,” A sanity check for ai-generated image detection”, ICLR 2025. \
> [2] Dhariwal, Prafulla, and Alexander Nichol. "Diffusion models beat gans on image synthesis." Advances in neural information processing systems 34 (2021): 8780-8794. \
> [3] Black Forest Labs, FLUX.1-dev, “https://github.com/black-forest-labs/flux”, 2024. \
> [4] Esser, Patrick, et al., "Scaling rectified flow transformers for high-resolution image synthesis.", ICML 2024. \
> [5] Kang, Minguk, et al., "Scaling up gans for text-to-image synthesis.", CVPR 2023. \
> [6] Midjourney Inc., “https://huggingface.co/datasets/saq1b/midjourney-v6.1”, 2024. \
> [7] Zhu, Mingjian, et al., "Genimage: A million-scale benchmark for detecting ai-generated image.", NIPS 2023.\
> [8] Karageorgiou, Dimitrios, et al., "Any-resolution ai-generated image detection by spectral learning.", CVPR 2025.

---

> ### Author Response · Authors · 2025-11-22
> **Rebuttal by Authors [4/4]**
>
> > **[W4]** Due to the nature of the method, I would also be very interested in how often the model misclassifies real images. I would suggest to include ablation studies by taking real images from ImageNet, COCO, etc, and show the misclassification scores.\
> **[Q4]** Can the authors show performance specific to real images?
>
> **[Response]** We appreciate your suggestion regarding the real-world image detection scenarios.
>
> Following your suggestion, we measure the misclassification score using real images from ImageNet, COCO [1], and RAISE-1k. Here, we define the misclassification score as "how often the detector incorrectly classifies real images as AI-generated" (please let us know if you intended a different metric). A real image is considered misclassified when its anomaly score exceeds a threshold. We determine this threshold using Youden's J-statistic [2], which maximizes balanced accuracy between avoiding false positives on real images and detecting generated images. For a reliable evaluation, we adopt 10-fold cross-validation, using 9 splits for threshold selection and 1 split for evaluating misclassification on real images.
>
> |  | ImageNet | COCO | RAISE-1k | Average |
> | :---- | :---- | :---- | :---- | :---- |
> | RIGID | 0.2446 | 0.3359 | 0.4190 | 0.3331 |
> | MINDER | 0.4073 | 0.5918 | 0.2314 | 0.4101 |
> | AEROBLADE | _0.2226_ | **0.1906** | _0.1308_ | _0.1813_ |
> | Manifold Bias | 0.3031 | 0.2715 | 0.3444 | 0.3063 |
> | SpAN (ours) | **0.1629** | _0.1958_ | **0.0956** | **0.1514** |
>
> As shown in the table, SpAN demonstrates robustness across the three real-world datasets, achieving the lowest average misclassification score (0.1514). We believe this result further highlights the generalizability of our method not only across diverse generator models but also across real-image distributions.
>
> **References**\
> [1] Lin, Tsung-Yi, et al., "Microsoft coco: Common objects in context.", ECCV, 2014. \
> [2] Ruopp, Marcus D., et al., "Youden Index and optimal cut‐point estimated from observations affected by a lower limit of detection.", Biometrical Journal: Journal of Mathematical Methods in Biosciences, 2008.

---

### Official Review · Reviewer_zN8D · 2025-10-30

**Soundness:** 2
**Presentation:** 1
**Contribution:** 2
**Rating:** 2
**Confidence:** 4

**Summary:**

The paper proposes SpAN, a training-free AIGI detection method based on spectral analysis. It observes that upsampling in generative models (e.g., transposed convolutions) introduces persistent spectral artifacts at axial Nyquist frequencies, even in photorealistic images. SpAN detects these via:

- Power calibration using azimuthal integration of high-frequency power to mitigate content bias,

- Autoencoder reconstruction to amplify residual artifacts and compute discrepancy in calibrated power between original and reconstructed images.

Experiments on Synthbuster and GenImage show SpAN significantly outperforms other training-free methods.

**Strengths:**

- SpAN requires no labeled data, no retraining, and outperforms other training-free baselines across proprietary and open-source models.

- The method is straightforward, making it easy to reproduce.

**Weaknesses:**

- The authors claim to be the first to directly leverage spectral-domain information in the Fourier space as a metric for training-free AIGI detection. However, utilizing frequency information to detect generated images has been extensively studied[1, 2, 3]. I believe these works need to be highlighted in the paper, emphasizing the differences between the proposed method and those methods.

- The empirical evidence presented (e.g., Figure 3(b)) suggests that the proposed spectral features, in isolation, possess limited discriminative power. This finding raises concerns about the centrality of the paper's claimed contribution, as the method's overall efficacy appears to be heavily reliant on the subsequent autoencoder component. The role of autoencoder requires further clarification. Furthermore, using an autoencoder incurs significant additional computational overhead.

- The paper's empirical validation is notably constrained. The omission of several key baseline detectors [1, 3, 4, 5] and the exclusion of diverse, large-scale datasets (e.g., DiffusionForensis, AIGCDetectBenchmark, DRCT2M, Chameleon) preclude a comprehensive comparative analysis, making it difficult to robustly assess the proposed method's performance and generalizability against the current state-of-the-art.

- The omission of an appendix is a notable concern. This absence prevents the authors from providing crucial supplementary information, such as experimental details.

Reference:

[1]: Thinking in frequency: Face forgery detection by mining frequency-aware clues

[2]: Frequency-aware deepfake detection: Improving generalizability through frequency space domain learning

[3]: Rethinking the Up-Sampling Operations in CNN-based Generative Network for Generalizable Deepfake Detection

[4]: Towards Universal Fake Image Detectors that Generalize Across Generative Models

[5]:  Manifold induced biases for zero-shot and few-shot detection of generated images

**Questions:**

Please see weakness.

---

> ### Author Response · Authors · 2025-11-21
> **Rebuttal by Authors [1/2]**
>
> Dear Reviewer zN8D,
>
> We are grateful for the reviewer‘s insightful feedback. We address each comment individually in the following responses.
>
> ---
>
> > **[W1]** The authors claim to be the first to directly leverage spectral-domain information in the Fourier space as a metric for training-free AIGI detection. However, utilizing frequency information to detect generated images has been extensively studied[1, 2, 3]. I believe these works need to be highlighted in the paper, emphasizing the differences between the proposed method and those methods.
>
> **[Response 1]**
> SpAN is the first to introduce an emerging spectral artifact that could be detected *without model training* via observing PSD near Nyquist frequency, rather than relying on learned frequency distribution. This could be an important step for detecting images on unseen or limited data, where training-based AIGI detectors are known to suffer.
>
> To support the necessity of training-free AIGI and alleviate your concerns, we have conducted cross-domain evaluation while including FreqNet [2] (AAAI, 2024) and SPAI [6] (CVPR, 2025) as the frequency-aware training-based baselines. Also, we added two training-free based methods, RIGID [7] and MINDER [8].
>
> |  | FreqNet \[2\] | SPAI \[6\] | RIGID [7] | MINDER [8] | SpAN (ours) |
> | :---: | :---: | :---: | :---: | :---: | :---: |
> | AUROC | 0.604 | 0.478 | 0.725 | 0.365 | **0.758** |
>
> As shown in the above table, our method achieves superior performance on cross-domain evaluation where the dataset has not been exposed to both the pre-trained autoencoder of our SpAN and frequency-aware training-based detectors. This experiment demonstrates that the prior knowledge learned in the frequency domain for training-based approaches can be a burden when inferencing the data with domain shift. In contrast, our method detects the artifacts without explicit training, which emerges from the upsampling process of a generator, being less dependent on the domain shift of the data.
>
>
> We also clarify the difference between SpAN and mentioned works [1, 2, 3]. [1] and [2] investigate frequency distribution of generated images by decomposing the amplitude into several bins or highlighting the high-frequency ranges captured in GANs. While both methods utilize raw values from the spectral domain, SpAN utilizes calibrated value via referencing adjacent areas in the PSD, which acts like a normalizing factor. This makes SpAN not being too sensitive to specific models, as the absolute value of PSD, by itself, is known to vary significantly. Moreover, while [3] observes the artifact related to the upsampling process of generators, the method investigates the artifact in the pixel space, apart from the Fourier or other frequency domains, which is a significant difference from our method.
>
> Following your suggestion, we have added the cross-domain evaluation in Table 7 of our manuscript, and included [2] in the ‘Introduction’ and ‘Related Works’ section to emphasize the difference of our method from previous frequency-aware training-based detectors. We thank you for giving us a chance to clarify, and hopefully could differentiate our method from previous works.
>
>
> **References** \
> [1]: Thinking in frequency: Face forgery detection by mining frequency-aware clues. \
> [2]: Frequency-aware deepfake detection: Improving generalizability through frequency space domain learning, AAAI 2024. \
> [3]: Rethinking the Up-Sampling Operations in CNN-based Generative Network for Generalizable Deepfake Detection. \
> [4]: Towards Universal Fake Image Detectors that Generalize Across Generative Models. \
> [5]: Manifold induced biases for zero-shot and few-shot detection of generated images, ICLR 2025. \
> [6]: Any-resolution ai-generated image detection by spectral learning, CVPR 2025. \
> [7]: Rigid: A training-free and model-agnostic framework for robust ai-generated image detection., arXiv preprint, 2024. \
> [8]: Understanding and improving training-free ai-generated image detections with vision foundation models., arXiv preprint, 2024.

---

> ### Author Response · Authors · 2025-11-21
> **Rebuttal by Authors [2/2]**
>
> > **[W2]** The empirical evidence presented (e.g., Figure 3(b)) suggests that the proposed spectral features, in isolation, possess limited discriminative power. This finding raises concerns about the centrality of the paper's claimed contribution, as the method's overall efficacy appears to be heavily reliant on the subsequent autoencoder component. The role of autoencoder requires further clarification. Furthermore, using an autoencoder incurs significant additional computational overhead.
>
> **[Response 2]**
> In SpAN, the use of an autoencoder should be considered as one of the main components in our framework, rather than considering the artifact as an isolated feature. The role of the autoencoder is to ‘inject the artifact near Nyquist frequency via reconstruction’,  the artifact which would not have existed in raw, real-world images. Therefore, instead of viewing the autoencoder as an extra component, the autoencoder is essential, which enables SpAN to compare the difference of PSD before and after the reconstruction process.
>
> Secondly, we respectfully disagree with the suggested weakness regarding the computation overhead. Other training-free based approaches such as AEROBLADE [1] and Manifold Bias [2] (included in our baselines) also utilizes autoencoder to reconstruct the original image, similar to SpAN. Also, considering that the autoencoder is only used for inference without training, the computation issue is considered to be negligible than training-based AIGI detection models as well.
>
> **References**
>
> [1] Ricker, et al., "Aeroblade: Training-free detection of latent diffusion images using autoencoder reconstruction error.", CVPR 2024. \
> [2] Brokman, Jonathan, et al., "Manifold induced biases for zero-shot and few-shot detection of generated images.", ICLR 2025.
>
>  ---
>
> > **[W3]** The paper's empirical validation is notably constrained. The omission of several key baseline detectors [1, 3, 4, 5] and the exclusion of diverse, large-scale datasets (e.g., DiffusionForensis, AIGCDetectBenchmark, DRCT2M, Chameleon) preclude a comprehensive comparative analysis, making it difficult to robustly assess the proposed method's performance and generalizability against the current state-of-the-art.
>
> **[Response 3]**
> Following your suggestion, we have conducted empirical validation on the Chameleon benchmark, as shown in the below table:
>
> |  | RIGID | MINDER | AEROBLADE | Manifold Bias | SpAN (ours) |
> | :---- | :---- | :---- | :---- | :---- | :---- |
> | AUROC | 0.566 | 0.480 | 0.589 | 0.655 | **0.756** |
>
> SpAN achieves the best performance in Chameleon benchmark, outperforming current state-of-the-art by **+0.101** points in AUROC. Note that Chameleon is considered a ‘hard’ benchmark in AIGI detection task, as human annotators additionally refined the dataset by leaving generated images that look deceptively realistic. SpAN can preserve its performance to these datasets since our method focuses on upsampling artifacts rather than relying on the complexity or semantic information of the image.
>
> Please note that Manifold Bias [5] is already included in our baselines, and SpAN outperforms the method by a large margin across all the three benchmarks (Synthbuster, GenImage, Chameleon). Moreover, we did not include [1, 3, 4] as these methods are ‘training-based’ AIGI detection models, which may not directly fit to our baselines.
>
> We also have included these additional benchmark results in the revised version (see Table 3).
>
> **References**
>
> [1]: Thinking in frequency: Face forgery detection by mining frequency-aware clues \
> [3]: Rethinking the Up-Sampling Operations in CNN-based Generative Network for Generalizable Deepfake Detection \
> [4]: Towards Universal Fake Image Detectors that Generalize Across Generative Models \
> [5]: Manifold induced biases for zero-shot and few-shot detection of generated images
>
> ---
>
> > **[W4]** The omission of an appendix is a notable concern. This absence prevents the authors from providing crucial supplementary information, such as experimental details.
>
> **[Response 4]**
> To resolve your concerns, we have included more experimental details (e.g., pseudo code of our method, further implementation details, and information about benchmarks) in the Appendix A, B, and C. We expect that the provided details could be beneficial for understanding our method better. Thank you for pointing this out.

---

### Official Review · Reviewer_uwfV · 2025-11-02

**Soundness:** 2
**Presentation:** 3
**Contribution:** 2
**Rating:** 4
**Confidence:** 3

**Summary:**

This paper presents a training-free AI generated image detector, which is based on frequency analysis. To achieve robust detection with the training-free model, this paper presents power calibration and applies auto-encoder to reconstruct a testing image which reveals different distributions on the spectral artifacts. To show the effectiveness of the proposed method, this paper conducts experiments on Synthbuster and GenImage datasets. Overall, I think this paper is not novel enough to publish in ICLR. As well known, the spectrum artifacts are explored a lot and many variations are made on the frequency analysis. Furthermore, as mentioned in this paper, the key observation is  upsampling operations in generative models inevitably introduce spectral artifacts, which remain most pronounced in the frequency domain. This is revealed by many previous work, in particular in GAN-based image generation. The diffusion model adds random noises on the image, and then the up-convolution issue is not clear. I vote for marginally below the acceptance threshold for this paper.

**Strengths:**

1. This paper proposed a simple method for AIGI detection. Overall, this paper is easy to follow. Ablation study is also provided.

2. To improve the robustness of the pipeline, a reconstruction process is applied. The proposed method shows better results than RIGID, MINDER, AEROBLADE, and Manifold Bias.

**Weaknesses:**

1. The idea is not novel enough. As well known, the spectrum artifacts are explored a lot and many variations are made on the frequency analysis. Furthermore, as mentioned in this paper, the key observation is  upsampling operations in generative models inevitably introduce spectral artifacts, which remain most pronounced in the frequency domain. This is revealed by many previous work, in particular in GAN-based image generation.

2. Cross-domain detection results will be helpful to improve this paper. As this paper mentioned that generalization is a problem for the training-based method, this paper needs to show more studies on cross-domain detection. As the different images are used to train the auto-encoder, the model still performs differently.

**Questions:**

Distribution gap is commonly existing in frequency domain, for example DCT, FFT domains. Why is Nyquist frequency applied in this work? What are the advantages of Nyquist frequencies compared with others?

---

> ### Author Response · Authors · 2025-11-21
> **Rebuttal by Authors [1/2]**
>
> Dear Reviewer uwfV,
>
> We sincerely thank you for your helpful feedback and insightful comments. We address your comments and questions below.
>
> ---
>
> > **[W1]** The idea is not novel enough. As well known, the spectrum artifacts are explored a lot and many variations are made on the frequency analysis.
>
> **[Response 1]**
> We respectfully disagree with the reviewer’s opinion. Our work (1) proposes ‘training-free’ method with appropriate calibration techniques and (2) suggests a different way to view PSD, focusing on the spatial locality as detailed below:
>
> 1. Our method, SpAN, is the first to directly utilize the emerging spectral artifacts in the Fourier domain in the **“training-free”** setting, while generalizing to various generator types (please refer to tables 1,2,3 in the manuscript and our experiment results in the **[Response 3]** to reviewer ‘Q8Fx’). We also propose a novel frequency-based calibration with image reconstruction (Section 3.2~3.3), which can amplify artifact signals, enabling us to detect spectral artifacts without explicit training. Please note that previous frequency-aware AIGI detection approaches are _training-based_, which requires learning frequency distributions of generative images. Therefore, the training data is necessary in their approaches, whereas SpAN does not.
>
> 2. Our frequency analysis differs from the previous analyses in that we suggest _specific regions of PSD_ (i.e., near Nyquist frequency) to investigate, while the previous analyses have mainly focused on PSD from a global perspective. However, we point out that if the PSD goes through appropriate power calibration considering the spatial locality of PSD (i.e., investigating PSD near the Nyquist frequency), spectral artifacts near Nyquist frequencies can be distinctive enough, even by a training-free manner.
>
> ---
>
> > **[W2]** Cross-domain detection results will be helpful to improve this paper. As this paper mentioned that generalization is a problem for the training-based method, this paper needs to show more studies on cross-domain detection. As the different images are used to train the auto-encoder, the model still performs differently.
>
> **[Response 2]** :
> Thank you for your insightful comment. Following your suggestion, we have conducted an evaluation on cross-domain (art domain) detection, where both the pretrained auto-encoder of our SpAN and existing training-based methods are not exposed to. For baselines, we selected two recent frequency-aware training-based methods, FreqNet [1] (AAAI, 2024), and SPAI [2] (CVPR, 2025). Also, we added two training-free methods, RIGID [3] and MINDER [4].
>
> |  | FreqNet \[1\] | SPAI \[2\] | RIGID \[3\] | MINDER \[4\] | SpAN (ours) |
> | :---: | :---: | :---: | :---: | :---: | :---: |
> | AUROC | 0.604 | 0.478 | 0.725 | 0.365 | **0.758** |
>
> These results demonstrate the superior generalization ability of our method in cross-domain evaluation over training-based approaches, supporting the limitation of training-based approaches of performance degradation on unseen datasets. This may be attributed to the fact that training-based approaches can suffer from the domain gap, such as differences of PSD per generator [5] or color distribution [6].
>
> However, our training-free approach can remain relatively robust, as one of the roles of the auto-encoder in our SpAN is to inject spectral artifacts at the reconstructed image. Therefore, even though there exists a domain gap between the dataset used to pretrain the auto-encoder and the test dataset, the artifact would likely emerge as the artifact stems from the upsampling operations, rather than from the specific context of the trained dataset.
>
> We appreciate the reviewer’s suggestion to include cross-domain dataset evaluation that could improve our manuscript. We have included the experiments and this discussion in the revised manuscript (see Section 4.4 and Table 7).
>
> **References**
>
> [1] Tan, Chuangchuang, et al., "Frequency-aware deepfake detection: Improving generalizability through frequency space domain learning.", AAAI 2024. \
> [2] Karageorgiou, Dimitrios, et al., "Any-resolution ai-generated image detection by spectral learning.", CVPR 2025. \
> [3] He, Zhiyuan, et al., "Rigid: A training-free and model-agnostic framework for robust ai-generated image detection.", arXiv preprint (2024). \
> [4] Tsai, Chung-Ting, et al. "Understanding and improving training-free ai-generated image detections with vision foundation models." arXiv preprint (2024). \
> [5] Corvi, Riccardo, et al., "On the detection of synthetic images generated by diffusion models”, ICASSP IEEE, 2023. \
> [6] Jia, Zexi, et al., “Secret Lies in Color: Enhancing AI-Generated Images Detection with Color Distribution Analysis”, CVPR 2025.

---

> ### Author Response · Authors · 2025-11-21
> **Rebuttal by Authors [2/2]**
>
> > **[Q1]** Distribution gap is commonly existing in frequency domain, for example DCT, FFT domains. Why is Nyquist frequency applied in this work? What are the advantages of Nyquist frequencies compared with others?
>
>
> **[Response]** \
> **[Q1-1]**. Why is the Nyquist frequency applied in this work? \
> Spectral artifacts near the axial Nyquist frequency are the points where generated images exhibit distinctive features in the PSD, compared to those of a real-world image. Real-world images are known to follow the power law of $1/f^{\beta}$ [1], exhibiting relatively small energy near the Nyquist frequency, in our context. However, for generated or reconstructed images, the replica of frequency energy gets added due to the upsampling process, making the Nyquist frequency a distinct region to discriminate between generated images from real-world images (see Figure 2a).
>
> **[Q1-2]** What are the advantages of Nyquist frequencies? \
> Firstly, Nyquist frequencies are less dependent on the type of generators. $c_\text{SpAN}$ investigates artifact arising regions, rather than being subject to a specific type of generative model. As you mentioned, there could exist a distribution gap in the frequency domain, especially in the mid or high-frequencies of the PSD, which varies across the dataset. However, Nyquist frequencies can commonly be captured since they originate from upsampling operations that are mostly included in recent generative models.
>
> Secondly, the artifact at Nyquist frequencies tends to remain over common perturbations, which is important when applying to real-world scenarios. Generally, artifacts in the frequency domain or pixel space are known to be prone to common postprocessing, if not explicitly trained with augmentations [2]. As demonstrated in the experiment of Section 4.3, SpAN exhibits robustness over perturbations compared to baselines. This suggests that the artifacts near the Nyquist frequency are noticeable enough, which is a reasonable choice within the PSD.
>
> **References**
>
> [1] Dzanic, et al., "Fourier spectrum discrepancies in deep network generated images.", Neurips 2020. \
> [2] Frank, Joel, et al., "Leveraging frequency analysis for deep fake image recognition.", PMLR 2020.

---

> ### Author Response · Authors · 2025-12-03
> **Rebuttal by Authors [2-1/2]**
>
> > **[Q1]** Distribution gap is commonly existing in frequency domain, for example DCT, FFT domains. Why is Nyquist frequency applied in this work? What are the advantages of Nyquist frequencies compared with others?
>
> **[Additinal Response to Q1]** To further demonstrate the advantage of utilizing axial Nyquist frequency for training-free AIGI detection, we conducted ablation on the position of artifact investigating region by downscaling the original axial Nyquist frequency $f_N$ to $k \cdot f_N$, where $k \in $ {0.2, 0.4, 0.6, 0.8}, at the Synthbuster benchmark. For a fair comparison, the region for determining $P_{\text{ring}}(\cdot)$ is selected so that the frequency corresponding to the outer ring becomes identical to the scaled frequency, $k \cdot f_N$.
>
> |  | $f\_N$ | $0.8f\_N$ | $0.6f\_N$ | $0.4f\_N$ | $0.2f\_N$ |
> | :---: | :---: | :---: | :---: | :---: | :---: |
> | AUROC | **0.944** | 0.556 | 0.547 | 0.511 | 0.517 |
>
> As shown in the above table, even if the points are selected from the same axis (i.e., $(\pm (k \cdot f_N), 0), (0, \pm (k \cdot f_N))$, areas except near $f_N$ does not exhibit features that could identify generated images from real-world images. This highlights the advantage of use of Nyquist frequency, compared to other specific points within the power spectrum density.
>
> Moreover, while other previous works have reported artifacts in the spectral domain for generated images, our work has distinction in that we specify the 'partial region' that remains discriminative within the power spectrum density. We validated that this phenomenon is is not only confined to GANs and also could be observed at images from various generative models including Stable Diffusion, Midjourney, and FLUX.

---

### Author Response · Authors · 2025-11-21
**General Response**

Dear reviewers and AC,

We sincerely appreciate your valuable time and effort spent reviewing our manuscript.

As reviewers highlighted, we propose a simple, easily adaptable, and efficient training-free AI-generated image detection method (Reviewer zN8D, Q8Fx), with superior performance over baselines (Reviewer uwfV, zN8D). Also, our paper is also easy to follow (Reviewer uwfV).

We appreciate your constructive comments on our manuscript. In response to the comments, we have carefully revised and enhanced the manuscript with the following additional discussions and experiments:
- Add one more benchmark to validate our superior performance over baselines (Table 3)
- Add an additional experiment regarding cross-domain inference with discussion (Table 7, Section 4.4)
- Add pseudo-code of our method, further implementation details, and information about the evaluation benchmarks (Appendix A, B, C)
- Clarify descriptions and statements throughout our manuscript

These updates are temporarily highlighted in “blue” for your convenience to check.

We hope our response and revision sincerely address the reviewers’ concerns.

Thank you very much.

Best regards, \
Authors.

---

> ### Comment · Reviewer_uwfV · 2025-11-23
> **Final Decision - Reviewer uwfV**
>
> I appreciate the effort from the authors, however, I decide not to change my initial rating, after reading the rebuttal and comments from other reviewers.

---

> ### Author Response · Authors · 2025-11-24
> **Response to Reviewer uwfV**
>
> Dear Reviewer **uwfV**,
>
> We appreciate the reviewer's time and for revisiting our work during the rebuttal phase.
>
> However, since the final comment is brief and does not indicate which parts of our responses remain unconvincing, it is difficult for us to further address your concerns.
>
> If possible, could you clarify which aspects of our rebuttal did not sufficiently resolve your initial points? We believe that continuing this discussion could help further improve our work.
>
> Thank you again for your time and thoughtful review.
>
> Best regards, \
> Authors

---

### Author Response · Authors · 2025-11-26
**A Gentle Reminder to AC and Reviewers**

Dear AC and Reviewers,

We hope this message finds you well. We are writing to kindly follow up regarding our rebuttal.

We have made a sincere effort to address the reviewers' concerns through detailed clarifications, additional experiments, and further analysis. We believe these additions meaningfully strengthen our paper and help clarify important points.

If there are any remaining concerns or unresolved issues, please feel free to let us know. We would be happy to provide further clarification.

Thank you again for your time and valuable feedback.

Best regards, \
Authors

---

### Meta-Review · Area_Chair_tp2q · 2026-01-07

**Summary:**

Across three reviews, the submission is viewed as a simple, training-free frequency-based detector with strong empirical results on the reported benchmarks and generally clear, reproducible presentation. The central concerns driving the decision are (i) limited novelty/positioning vs. prior frequency-based detection work, (ii) whether the core contribution is the proposed spectral cue vs. the autoencoder reconstruction component (and associated compute), and (iii) insufficient breadth of evaluation/baselines in the original submission (cross-domain, additional datasets, real-image false positives, and broader generator coverage).

Despite meaningful rebuttal additions, the record still leaves material uncertainty about novelty/positioning and the core technical contribution (spectral cue vs. reconstruction), and the original request for more comprehensive validation is only partially addressed. With one firm reject and only one confirmed post-rebuttal stance (unchanged), the evidence is insufficient to recommend acceptance.

**Reviewer Concerns:**

Partially addressed concerns:
- Authors added a cross-domain evaluation and argue training-free robustness vs. training-based frequency methods.
- Authors provided motivation and an added ablation over frequency-region choices.
- Authors added at least one additional benchmark (notably “Chameleon”) and more comparisons, but the original concern about comprehensiveness remains only partially resolved (several requested datasets/baselines were not incorporated, and some exclusions are justified as “training-based”).
- Authors clarified that reconstruction is integral to the method and argued overhead is comparable to prior training-free approaches, but the critique that the spectral cue alone is weak and that the method’s effectiveness hinges on the reconstruction step remains a legitimate interpretive concern.
- Authors argue the method targets artifacts from upsampling broadly and cite results/analyses on generators without explicit transposed convolutions.
- Authors provided additional targeted analyses (hard subset, inpainting behavior, and real-image misclassification rates). However, inpainting remains a weaker regime per their own reported results, suggesting an acknowledged limitation.

**Reviewer Scores:**

- uwfV: 4 → 4 (no change) (explicitly stated by the reviewer).
- zN8D: 2 → 2–3 (likely +0 to +1): added benchmarks/appendix and clearer positioning could soften the “insufficient validation” complaint, but novelty/centrality concerns plausibly keep them in reject territory.
- Q8Fx: 4 → 4–5 (likely +0 to +1): most questions were directly answered with added experiments/analyses; remaining weaknesses (e.g., inpainting sensitivity) could still prevent a clear move above threshold.

---

### Decision · Program_Chairs · 2026-01-26

Reject